# Can the Duration of In-Hospital Ventilation in Patients with Sepsis Help Predict Long-Term Survival?

**DOI:** 10.3390/jcm11205995

**Published:** 2022-10-11

**Authors:** Moti Klein, Adir Israeli, Lior Hassan, Yair Binyamin, Dmitry Frank, Matthew Boyko, Victor Novack, Amit Frenkel

**Affiliations:** 1General Intensive Care Unit, Soroka University Medical Center, the Faculty of Health Sciences, Ben-Gurion University of the Negev, Beer-Sheva 84101, Israel; 2The Joyce and Irving Goldman Medical School, Faculty of Health Sciences, Ben-Gurion University of the Negev, Beer-Sheva 8410101, Israel; 3Clinical Research Center, Soroka University Medical Center, the Faculty of Health Sciences, Ben-Gurion University of the Negev, Beer-Sheva 84101, Israel; 4Department of Anesthesiology, Soroka University Medical Center, the Faculty of Health Sciences, Ben-Gurion University of the Negev, Beer-Sheva 84101, Israel; 5Anesthesia, Critical Care and Pain Medicine, Beth Israel Deaconess Medical Center, Harvard Medical School, Boston, MA 02215, USA

**Keywords:** intensive care unit, prolonged ventilation, mortality, sepsis

## Abstract

Mechanical ventilation is a cornerstone in the treatment of critical illness, especially sepsis. Prolonged mechanical ventilation, for a duration exceeding 21 days, is associated with higher rates of in-hospital and post-discharge mortality. Our aim was to assess the association between in-hospital ventilation duration and long-term life expectancy in patients ventilated in intensive care units specifically due to sepsis of any origin. We conducted a population-based retrospective cohort study of adults hospitalized in a general intensive care unit for 24 h or more during 2007–2017, who were diagnosed with sepsis or septic shock, treated with invasive mechanical ventilation for a maximum of 60 days and survived hospitalization. The primary exposure was the length of invasive mechanical ventilation. In an adjusted multivariable regression model, survival rates at 1, 2, 3 and 4 years post-hospitalization did not differ significantly between patients who were ventilated for 3–8 days (n = 169), 9–21 days (n = 160) or 22–60 days (n = 170), and those who were ventilated for 1–2 days (n = 192). We concluded that the duration of in-hospital ventilation in patients with sepsis cannot serve as a predictor for long-term survival. Thus, the duration of ventilation in itself should not guide the level of care in ventilated patients with sepsis.

## 1. Introduction

Mechanical ventilation is a cornerstone in the treatment of critically ill patients, especially those with sepsis, and has been identified as a key predictor of mortality in this population. Aziz et al. (2017) [1] identified low platelet count, elevated serum levels of C-reactive protein, and the need for invasive mechanical ventilation as independent predictors of mortality among adults with sepsis who were treated in critical care units. In the same context, Pittet et al. [2] published a model, for bedside use, aimed to predict the mortality from sepsis in patients treated in critical care units. Their model showed that independent predictors of mortality were previous antibiotic therapy, hypothermia, onset-of-sepsis APACHE II score and the requirement for mechanical ventilation.

A number of studies investigated the correlation specifically between the duration of mechanical ventilation and mortality. When prolonged mechanical ventilation (PMV) was defined as mechanical ventilation (MV) for a duration of more than 21 days [3], PMV was found to be associated with higher rates of in-hospital and post-discharge mortality, decreased functional capacity, and poor quality of life after intensive care [4,5,6,7,8]. One of the largest studies that examined the correlation between PMV and life expectancy is a systematic review and meta-analysis of 124 studies from 16 countries, published by Damuth et al. in 2015 [6]. Their primary outcome was mortality at 1 year; the secondary outcome was in-hospital mortality. Pooling data of 39 studies that reported mortality at 1 year yielded a rate of 59% (95% CI 56–62). Among the 29 high-quality studies, the pooled mortality at 1 year was 62% (95% CI 57–67). The pooled mortality at hospital discharge was 29% (95% CI 26–32). The authors concluded that although a high proportion of patients survived to hospital discharge, fewer than half the patients survived beyond 1 year. 

To the best of our knowledge, none of the studies that examined life expectancy in patients who underwent PMV performed an analysis focusing specifically on patients who were ventilated due to sepsis, which is one of the leading indications for MV in patients hospitalized in critical care units [9]. Furthermore, none of the studies analyzed life expectancy after discharge from the hospital, according to the duration of in-hospital ventilation for this specific population. 

Thus, the primary aim of the current study was to assess the association between in-hospital ventilation duration and long-term life expectancy in patients who were ventilated in the intensive care unit (ICU) specifically due to sepsis of any origin. 

## 2. Methods

### 2.1. Study Population

We conducted a population-based retrospective cohort study at Soroka University Medical Center, a tertiary care medical center that serves as the only regional hospital in southern Israel (Beer-Sheva vicinity, estimated population of 1,000,000). We included patients above age 18 years who were hospitalized between January 2007 and December 2017 in the general ICU for more than 24 h, and were diagnosed with sepsis or septic shock, treated with invasive mechanical ventilation (IMV) and survived hospitalization. We included only patients who were ventilated during hospitalization in the general ICU for a maximum of 60 days. 

Figure 1 presents the patient flow chart, according to the study inclusion criteria.

### 2.2. Primary Exposure and Outcome Assessment—Study Design 

The primary exposure was the length of IMV. Patients were classified into four groups according to the duration of IMV: 1–2, 3–8, 9–21 and 22–60 days. The categories were determined to stratify the population to quartiles, as much as possible. The severity of sepsis was assessed based on the patients’ Sequential Organ Failure Assessment score (SOFA). For each patient, the SOFA score was calculated based on the closest data to the beginning of IMV in the general ICU. In a sub-analysis, patients were classified into five groups, according to their SOFA scores: 0–6, 7–8, 9–10, 11–12 and 13–19. The categories were determined to stratify the population to quintiles, as much as possible.

The follow-up period after discharge from the hospital was set at a maximum of four years. Data from sources outside the hospital were lacking; therefore, the only information collected following discharge was survival. Survival was examined at one-, two-, three- and four-year landmarks; and mortality was referred to as death from any cause. The overall survival was defined 4 years from hospital discharge. 

### 2.3. Statistical Analysis

Descriptive statistics are presented using summary tables. Continuous variables are described by means and standard deviations, and categorical variables by numbers and percentages. Comparisons between groups are presented by 95% confidence intervals and/or *p*-values. Percentages are rounded to one decimal place.

For continuous variables, the *t*-test was used for normally distributed variables. Differences between dichotomous variables were examined using the chi square test.

Non-parametric variables (such as ventilation length, hospitalization length, time in the general ICU and SOFA score at the start of ventilation) were tested using the Mann–Whitney and the Kruskal–Wallis test for variables not normally distributed, as appropriate. As a single variable test for patients’ survival, we used the Kaplan–Meier test. In multivariable modelling, variables were selected according to clinical and statistical significance: first, baseline clinical characteristics and age; then SOFA groups. The variables were introduced into proportional hazards regression (COX regression model). Ventilation groups (VG) and SOFA score groups were analyzed as categorical variables. The reference groups for hazard ratios (HRs) were the first group in both cases. We examined correlations between variables before introducing them to the multivariable model. Variables with a correlation higher than 70% (according to the Pearson test) were not added to the model, to prevent biases. Variables that were statistically significant (*p* < 0.05) in the univariable analysis were introduced to the model. All the statistical tests and confidence intervals, as appropriate, were at =0.05 (two-sided) or =0.025 (1-sided). All the *p*-values reported were rounded to three decimal places. A two-sided *p* < 0.05 was considered statistically significant. All the analyses were performed using SPSS, version 26 (IBM, Armonk, NY, USA). 

## 3. Results

### 3.1. Study Population

The study included 691 patients stratified to four groups according to the number of days they received IMV: 1–2 (n = 192, 27.8%), 3–8 (n = 169, 24.5%), 9–21 (n = 160, 23.2%) and 22–60 (n = 170, 24.6%). Table 1 summarizes the characteristics of the study population according to the duration of IMV. The majority of patients in all the groups were males; the proportion of males was the greatest for the highest duration group (*p*-value = 0.029), and hypertension was the most prevalent comorbidity in all the groups. 

### 3.2. Hospitalization Data

Table 2 summarizes hospitalization data according to the duration of IMV. The lowest calculated median SOFA score was observed in the first VG, and was higher and equal in the three other VGs. The median ICU hospitalization was the shortest for the first VG (2 days, IQR 1–4), and the longest for VG 4 (31 days, IQR 26–38), PV < 0.001.

### 3.3. Survival

Table 3 presents post-hospitalization survival, using the Kaplan–Meier test. After one year of follow-up, VG1 had the highest survival rate (87%). The survival rate of VG2 was lower than that of all the other VGs, at each of the four assessment points. 

Figure 2 presents the survival rates during four years of follow up, according to Kaplan–Meier analysis. Survival rates were similar for VG1 and VG3 at four years follow-up, and for VG1 and VG4 at two years follow-up.

The durations of invasive mechanical ventilation were 1–2, 3–8, 9–21 and 22–60 days for groups 1–4, respectively.

### 3.4. Multivariable Regression Model

Table 4 depicts the results of the multivariable analysis for the association between ventilation duration and post-hospitalization survival. Age and diabetes mellitus were found to be associated with increased risk of mortality. The higher mortality rates in VG2, VG3 and VG4 compared with VG1 were not statistically significant, nor were the higher mortality rates in the SOFA groups 2, 3 and 4 compared with SOFA group 1. In a COX regression model, VG 4 showed the highest survival rate throughout the entire follow-up period, and VG 2 showed the lowest.

## 4. Discussion 

The main finding of our study was that the duration of in-hospital ventilation in patients with sepsis who survived hospitalization cannot serve as a predictor for long-term survival. We also report that two parameters related to survival were age (HR 1.03, 95%CI 1.02–1.04) and diabetes mellitus (HR 1.45, 95%CI 1.09–1.94). 

Any comparison of our findings to other research should consider the specific characteristics of our population, namely, patients with sepsis who were treated with MV for more than 24 h and who survived hospitalization. Hill et al. [10] described a population-based cohort of adults who received MV in an ICU in Ontario, Canada during 2002–2013. They examined the association between PMV and 1-year mortality among 6678 patients who underwent PMV for any reason and who survived hospitalization. Among hospital survivors, mortality was 16.6% after 1 year. This compares to 13.0%, 21.3%, 18.1% and 14.7% for ventilation groups 1–4, respectively, of our cohort. These findings roughly support the mortality rates that were found in our study, while differences in characteristics between the two populations may explain the varying mortality rates. In their study, there was also no significant difference in 5-year survival after discharge from hospitalization, between three groups of patients who were ventilated for less than 64 days (22–32, 33–46 and 47–64 days). Once again, this finding supports our main finding, namely that the duration of in-hospital ventilation cannot serve as a predictor for long-term survival in patients with sepsis. 

Another factor that should be considered in the interpretation of our results is the limited number of beds in the ICU of our hospital, thus dictating a “closed door policy”. This necessitates admitting to the ICU the patients whose chances of survival are initially better. Assuming that those admitted have relatively good chances of survival, if they also survive the hospitalization, the duration of the ventilation that they underwent may not be a major influence on their long-term survival.

Interestingly, at all assessment points, the survival rate was the highest for the group of patients with the longest duration of ventilation, who also had the longest duration of hospitalization. Moreover, the survival rates were the lowest for the patients with the shortest duration of ventilation, who also had the shortest duration of hospitalization. We assume that these findings may be explained by the younger mean age (49.9 ± 20.8 years) and the lower proportions of chronic diseases among the patients with the longest ventilation duration compared with the other groups. Similarly, for the group of patients with the second shortest ventilation duration, and the worst survival, the mean age was the oldest (56.3 ± 19.2) and the proportions with chronic diseases the highest. 

Our study also found that SOFA score cannot predict long-term survival in the population examined. This finding is expected and similar to former studies. The SOFA score, which was designed as a research tool for categorizing patients based on their risk of death [11], provides valuable prognostic information for in-hospital survival when applied to patients with sepsis [12], but apparently not for out-of-hospital, long-term survival. 

Our study has a number of limitations. The first of them is the single-center design. Second, the lack of information regarding causes of mortality and mortality within specific diseases groups precluded analyzing data or reaching conclusions regarding differences in cause-specific mortality according to the duration of ventilation. Similarly, information was lacking regarding the sources of sepsis and indications for mechanical ventilation; this would have contributed to understanding the characteristics of the patients included and differences between them according to the duration of ventilation. Additionally, we do not know whether the patients were released from the hospital to their homes or to nursing institutions, nor the cognitive, respiratory and functional status of the patients at discharge. A key strength of the present study is the selection of the population. On the one hand, the selective inclusion of only the patients who survived the hospitalization distorts the real mortality data; on the other hand, it enables focusing on the survivors. To the best of our knowledge, this is the first published study in which the landmark analysis included only patients who were ventilated in a critical care unit specifically due to sepsis, and who survived the hospitalization. The findings presented here can contribute to developing understanding of the long-term physiological significance of the duration of ventilation in this population.

## 5. Conclusions

We suggest that the duration of in-hospital ventilation in patients with sepsis who survived hospitalization does not affect mortality during the four subsequent years. This conclusion may have implications for intensive care physicians who are required to decide the level of care for patients on a daily basis. It seems that even patients who are ventilated for long periods of time in ICUs due to sepsis should not be expected to have shorter survival in the following years, provided that they survived the hospitalization. Therefore, the duration of ventilation in itself should not guide the level of care in this population.

## Figures and Tables

**Figure 1 jcm-11-05995-f001:**
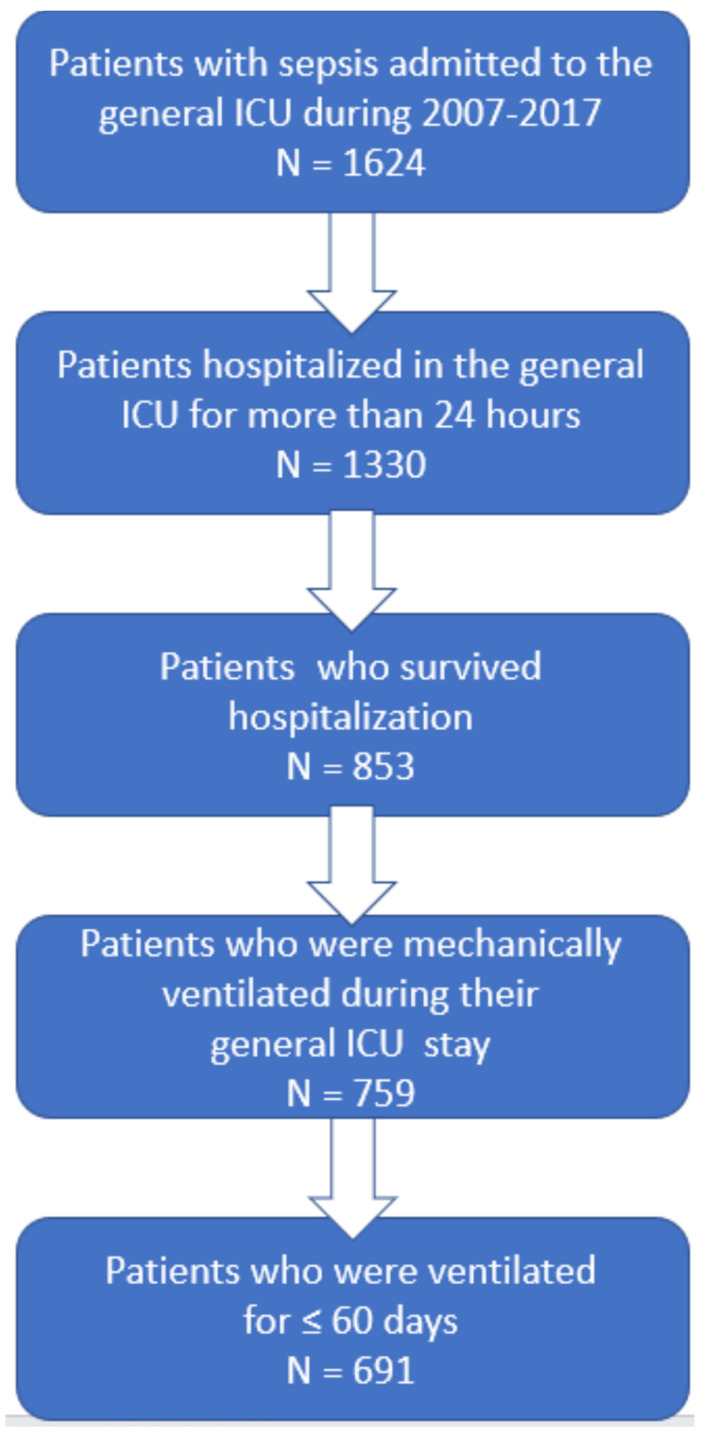
The patient flow chart, according to the study inclusion criteria.

**Figure 2 jcm-11-05995-f002:**
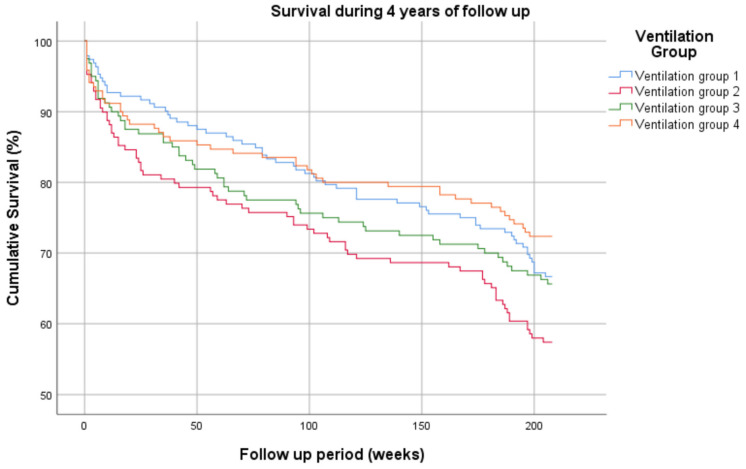
Four-year survival according to Kaplan–Meier analysis for four groups that differed by the duration of invasive mechanical ventilation.

**Table 1 jcm-11-05995-t001:** Demographic and clinical characteristics of the patients according to the duration of invasive mechanical ventilation.

	Group 1 (n = 192)	Group 2 (n = 169)	Group 3 (n = 160)	Group 4 (n = 170)	*p* Value
Age (Mean ± SD)	54.97 (19.95)	56.34 (19.23)	55.31 (19.23)	49.94 (20.78)	0.015
Male gender	108 (56.3%)	108 (63.9%)	94 (58.8%)	120 (70.6%)	0.029
Hypertension	62 (32.3%)	52 (30.8%)	43 (26.9%)	39 (22.9%)	0.206
CVA	7 (3.6%)	5 (3%)	5 (3.1%)	4 (2.4%)	0.915
Diabetes mellitus	50 (26%)	48 (28.4%)	34 (21.3%)	27 (15.9%)	0.030
Ischemic heart disease	22 (11.5%)	13 (7.7%)	8 (5%)	7 (4.1%)	0.032
Chronic kidney disease	10 (5.2%)	11 (6.5%)	10 (6.3%)	4 (2.4%)	0.283
Solid tumor	3 (1.6%)	0 (0%)	1 (0.6%)	0 (0%)	0.157
Smoking	53 (27.6%)	42 (24.9%)	38 (23.8%)	34 (20%)	0.303
COPD	22 (11.5%)	18 (10.7%)	6 (3.8%)	13 (7.6%)	0.047

The durations of invasive mechanical ventilation were 1–2, 3–8, 9–21 and 22–60 days for groups 1–4, respectively. CVA, cerebrovascular accident. COPD Chronic Obstructive Pulmonary Disease.

**Table 2 jcm-11-05995-t002:** Hospitalization data according to the duration of invasive mechanical ventilation.

	Group 1 (n = 192)	Group 2 (n = 169)	Group 3 (n = 160)	Group 4 (n = 170)	*p* Value
SOFA score—Median (IQR)	7 (5–8)	9 (7–11)	9 (7–11)	9 (7–11)	<0.001
ICU hospitalization duration (days) Median (IQR)	2 (1–4)	8 (6–10)	16 (13–20)	31 (26.75–38.25)	<0.001
Invasive ventilation duration (days) Median (IQR)	1 (1–1.18)	5.17 (3.8–6.7)	13.7 (10.65–17.74)	36.58 (28.74–46.43)	<0.001
Use of tracheostomy	11 (5.7%)	15 (8.9%)	72 (45%)	161 (94.7%)	<0.001

The durations of invasive mechanical ventilation were 1–2, 3–8, 9–21 and 22–60 days for groups 1–4, respectively. SOFA, Sequential Organ Failure Assessment; IQR, interquartile range. Post hoc analysis. Dunn’s test. Dunn’s test for SOFA score between group 1 and 2; 1 and 3; 1 and 4, PV < 0.001. The rest are PV > 0.05. Dunn’s test for ICU days between each group PV < 0.001.

**Table 3 jcm-11-05995-t003:** The cumulative survival proportion according to the duration of invasive mechanical ventilation.

		Group 1 (n = 192)	Group 2 (n = 169)	Group 3 (n = 160)	Group 4 (n = 170)
1-year survival	Estimate (std.)	87.0% (0.024)	78.7% (0.031)	81.9% (0.03)	85.3% (0.027)
2-year survival		80.2% (0.029)	72.8% (0.034)	75.0% (0.034)	80.6% (0.30)
3-year survival		75.5% (0.031)	68.0% (0.036)	71.9% (0.036)	78.2% (0.032)
4-year survival		66.7% (0.034)	57.4% (0.038)	65.6% (0.038)	72.4% (0.034)
*p* values Kaplan–Meier test		0.035			

**Table 4 jcm-11-05995-t004:** Cox regression multivariable model results for factors associated with post-hospitalization survival.

	Hazard Ratio	95% Confidence Interval	*p* Value
Age (per year)	1.03	1.02–1.04	<0.001
Gender (male)	0.78	0.59–1.02	0.073
Diabetes mellitus	1.45	1.09–1.94	0.01
Ischemic heart disease	1.2	0.79–1.79	0.397
VG, when the reference is VG1 (1–2 ventilation days)			0.544
VG2 (3–8 ventilation days)	1.12	0.74–1.69	0.582
VG3 (9–21 ventilation days)	1.31	0.90–1.91	0.157
VG4 (22–60 ventilation days)	1.12	0.75–1.66	0.569
SOFA groups, when the reference is group 1 (SOFA scores 0–6)			0.362
SOFA group 2 (SOFA scores 7–8)	0.69	0.43–1.13	0.144
SOFA group 3 (SOFA scores 9–10)	0.74	0.47–1.18	0.209
SOFA group 4(SOFA scores 11–12)	0.99	0.63–1.54	0.975
SOFA group 5 (SOFA scores 13–19)	0.89	0.56–1.42	0.892

VG, ventilation group; SOFA, Sequential Organ Failure Assessment. VGs were introduced as categorial variables into the COX model; the reference group for the hazard ratios was the first group. SOFA groups were introduced as categorial variables into the COX model; the reference group for the hazard ratios was the first group.

## Data Availability

The data used in the analysis of this study are not publicly available due to national regulations, but are available from the corresponding author upon request and following the Ethics Committee approval.

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
