# Peer review of "Can the Duration of In-Hospital Ventilation in Patients with Sepsis Help Predict Long-Term Survival?"

_jcm, 2022, doi:10.3390/jcm11205995_

Round 1

Reviewer 1 Report

This retrospective study looks at the correlation between length of mechanical ventilation and long term survival in critically ill septic patients who survived their hospital admission. Patients were divided into 4 groups based on the duration of mechanical ventilation. The authors found no statistically significant difference between outcomes in the different groups, concluding that length of ventilation in this cohort does not predict long term outcome.

The paper is well written however I have several comments:

There may be a large clinical variability between the ventilation groups not addressed in the data provided, additional clinical data is required. External validity and extrapolation of these findings are limited when  information is missing.

1. Please provide information regarding the source of sepsis in the patient cohort (patients with pneumonia may be ventilated for longer) as well as the indication for ventilation (shock, respiratory failure, acidosis etc). Although these may not be critical prognostic factors for long term outcomes, they may add a better understanding of the type of patients studied and the difference between the groups.

2. What is the incidence of chronic obstructive or other chronic lung disease in your cohort? This  may effect ventilatory and survival outcomes.

3. Patients who are ventilated for more than 14-21 days may require a tracheostomy, prolonging their weaning time and occasionally reducing their rehabilitation reserve and functional independence. Please provide any relevant information

4. How many patients were discharged from hospital still mechanically ventilated? Also, patients who are discharged to chronic ventilation centers may have a survival advantage in terms of monitoring and provision of chronic critical care. As cognitive, respiratory and functional status at discharge and subsequently to that are not provided, the authors should address these issues, because looking only at long term survival is a partial assessment of outcome.

In any case, if data is unavailable, please address this in the methods, limitations and discussion.

5. Lines 208-210 - how is the difference in age in group 4 versus 2 explained? Any hypotheses?

6. Study limitations should address the bias caused by looking only at patients who survive their hospital admission. Patients who die in hospital after being ventilated for many days will be excluded. How does this impact study conclusions? Please discuss.

7. The authors should explain the added information provided by figure 3 and consider its necessity.

8. Line 216 should be "It is"

Reviewer 2 Report

Are you able to look at the mortality of specific groups of our patients, for example those with severe copd, dialysis dependent acute kidney injury 
